# Interaction Between Enhancers and Promoters in Chicken Genome

**DOI:** 10.3390/ijms262311407

**Published:** 2025-11-25

**Authors:** Valentina A. Grushina, Anastasia P. Filatova, Valeria S. Gagarina, Danila E. Prasolov, Fedor A. Kolpakov, Oleg A. Gusev, Sergey S. Pintus

**Affiliations:** 1Department of Computational Biology, Sirius University of Science and Technology, 354340 Sirius, Russia; grushina_v@bk.ru (V.A.G.); gagarina.vs@talantiuspeh.ru (V.S.G.); danila.prasolov01@gmail.com (D.E.P.); koplakov.fa@talantiuspeh.ru (F.A.K.); 2Department of Genetics and Cell Biology, Tomsk State University, 634050 Tomsk, Russia; anastasijafilatova01@yandex.ru; 3Regulatory Genomics Research Center, Institute of Fundamental Medicine and Biology, Kazan Federal University, 420008 Kazan, Russia; gaijin.ru@gmail.com; 4LIFT Center LLC, 121205 Moscow, Russia

**Keywords:** enhancers, promoters, gene regulation, CAGE, chicken

## Abstract

Gene expression from promoters is influenced by interactions with genomic enhancers located within the same topologically associating domain (TAD). Enhancer activity can be evaluated by measuring the transcriptional output of enhancer RNAs, and the CAGE methodology enables the simultaneous assessment of enhancer and promoter activities within a single experiment. In this study, we examined the correlation between gene and enhancer activities within individual TADs across multiple tissues in slow- and fast-growing chickens, and we assessed the biological significance of genes with promoters that are regulated by enhancers. Our analysis revealed a statistically significant association between gene expression levels and enhancer activity in all tissues examined. Notably, enhancer-mediated regulation appears to activate key pathways involved in transcriptional control and nucleic acid biosynthesis.

## 1. Introduction

Epigenetic regulation constitutes a complex level of genomic activity control, involving DNA and histone modifications, as well as changes in chromatin structure without altering the nucleotide sequence itself. These mechanisms ensure the spatial and temporal precision of gene expression and underpin processes such as cellular differentiation, responses to stress factors, and the development of phenotypic traits. A central component of epigenetic regulation is the interaction among DNA regulatory elements, including promoters, silencers, insulators, and notably enhancers.

Genomic enhancers are non-coding DNA regions capable of enhancing the transcription of target genes regardless of their position or orientation relative to the promoter. Unlike promoters, which initiate transcription, enhancers modulate expression levels, providing tissue-specific and developmental stage-specific regulation [1]. The enhancer function is mediated through their interaction with transcription factors and coactivators, leading to the formation of an active chromatin environment and facilitating the recruitment of the transcriptional machinery to the gene promoter.

Enhancers in chickens, as in mammals, are identified based on characteristic epigenetic features, including the presence of histone modifications H3K4me1 and H3K27ac, open chromatin regions detected by ATAC-seq or DNase-seq, and transcriptional activity reflected by non-coding enhancer RNAs [1]. Current studies have shown that enhancers are marked by specific epigenetic signatures such as acetylation of histone H3 at lysine 27 (H3K27ac) and monomethylation at lysine 4 (H3K4me1), while active promoters are predominantly marked by H3K4me3 [2,3,4]. These features enable the use of chromatin immunoprecipitation followed by sequencing (ChIP-seq) to identify and annotate potential enhancers across various cell types and tissues [2]. The application of ChIP-seq in chickens has facilitated the annotation of active enhancers under different tissue contexts and conditions [5].

The application of the CAGE method in chickens has enabled the precise identification of transcription start sites (TSSs) and the detection of clusters corresponding not only to promoters but also to active enhancers [6]. Additionally, hundreds of genomic regions in the chicken genome have been described that exhibit characteristics of enhancer transcripts—namely, low-level bidirectional transcription associated with H3K27ac and the absence of H3K4me3 [7]. These elements were found to be tissue-specific, occurring in the liver, kidney, brain, and intestine.

In this work, we performed computational analyses of statistical associations between enhacers and promoters within reach of their topologically associating domain (TAD).

## 2. Results

### 2.1. Similarity Between Erythrocyte and Fibroblast TADs and the Consensus TAD Set

We used the consensus overlap between the erythrocyte and fibroblast sets from the Ontogen databese [8] lifted over from the original galGal5 assembly version to the GRCg7b genome used in our work as the reference set of TAD intervals (see Appendix B). To validate the consensus intervals, we calculated the number of fibroblast-specific and erythrocyte-specific TADs in novel genomic coordinates (Table 1).

The number of tissue-specific TADs was surprisingly small, as only 3.3% of fibroblast TADs and 1.74% of erythrocyte TADs were tissue-specific, with the Jaccard index being rather close to 1. On the one hand, both erythrocytes and fibroblast derive from the same bone marrow progenitor cell. On the other hand, those cell types have totally different fates with respect to hematopoiesis. This notion allowed us to hypothesize that the consensus overlap set of TAD intervals between the Ontogen erythrocyte and fibroblast sets could be used as a reference for our six tissues under study. Although such suggestions may seem to be a long-range extension of the observation made in two hematopoietic cell types, it is known that a substantial number of TADs are conserved between tissues [9]. Thus, given the high conservation of the Ontogen TADs, we found it feasible to use the consensus set of overlapping intervals as a reference for further study.

### 2.2. Enhancer-Mediated Epigenetic Regulation of Promoter Activity

To interpret the contribution of enhancers to transcriptional regulation, particularly in the context of differences between fast- and slow-growing chickens, potential interactions between TSSs and enhancers were modeled based on structural chromatin organization. Since regulatory element interactions are predominantly confined within TAD boundaries [10], experimentally defined TAD regions for chicken, derived from the Hi-C data of blood cells and fibroblasts [8], were incorporated into the analysis. Despite tissue-specific differences, these TAD profiles show substantial overlap, enabling their use for reconstructing spatial contacts in the tissues studied.

Mapping TSSs and enhancers located within the same TAD enabled the construction of a potential interaction map comprising over 23 million pairs. However, this extensive dataset did not reflect growth-specific regulation. Therefore, pairs including only differentially expressed TSSs and enhancers between fast- and slow-growing chickens were filtered for each tissue separately. The resulting dataset reflected regulatory interactions that are potentially associated with the growth phenotype.

To enhance the reliability of the interaction analysis, TSS–enhancer pairs were cross-referenced with TAD maps derived from chicken blood cells and fibroblasts. In cases where a blood cell TAD overlapped multiple fibroblast TADs, these were merged into the consensus set of TADs (see Materials and Methods). This approach accounted for tissue-specific variability in TAD structures during the interpretation of spatial constraints on interactions and improved the accuracy of cis-regulatory interaction identification.

Analyses of the identified interactions revealed that most TADs contain multiple TSSs along with their corresponding multiple enhancers, consistent with current models of modular genetic regulation [11,12,13]. Enhancers displayed bidirectional regulatory effects, capable of both activating and repressing transcriptional activity, as evidenced by the mixed directions of expression changes.

Notably, comparisons of the expression change directions between enhancers and their associated TSSs showed a predominance of positive regulation in the context of growth (Figure 1). Specifically, most cases were characterized by either the activation of all differentially expressed enhancers and all differentially expressed promoters within a TAD (positive value of the logarithm of fold change in both enhancers and TSSs; interactions of the “+ +” type) or the downregulation of all differentially expressed promoters along with the upregulation of all differentially expressed enhancers within one TAD (negative log fold change in promoters and positive log fold change in ehancer interactions of the “− +” type; see Appendix B). This suggests that in fast-growing chickens, transcriptional regulation is largely driven by the activation of specific regulatory clusters functioning within individual TADs.

We identified TAD regions demonstrating coordinated regulatory effects (Figure 2). Within such TADs, all differentially exresssed promoters exhibited the same response (upregulation or downregulation) to enhancer activity. This pattern indicates the existence of regulatory modules that orchestrate gene expression within discrete functional units. Such modular regulation appears to play a critical role in tissue-specific processes, such as skeletal muscle growth and hepatic metabolic function. Importantly, the proportion of positive regulatory interactions remained stable even in domains characterized by a uniform enhancer response.

Interactions of the “− + ” type, characterized by increased enhancer activity concurrent with decreased TSS expression, were observed significantly less frequently but are of particular interest. Such inversely directed changes may reflect regulatory redistribution of contacts within a TAD, wherein an active enhancer switches from one promoter to another. This could involve switching between alternative promoters, the engagement of repressors or insulator elements, or the reorganization of sub-TAD structure in response to tissue-specific stimuli. These patterns may represent transient states typical of differentiation processes or adaptation to physiological stress [14,15].

In addition to identifying interacting TSS–enhancer pairs localized within the same TAD, we performed an analysis of transcription factors (TFs) and cofactors for which their expression was also altered under conditions modeling growth differences. The annotation of TFs and cofactors was conducted using the AnimalTFDB 4.0 database [16], which includes chicken transcriptional regulators. A predominance of positive regulation (+ + pattern) was observed across all tissues, particularly in the liver, brain, and kidneys, indicating the activation of transcriptional programs in fast-growing chickens (Table 2). These TFs and cofactors were localized within the same TADs as the differentially expressed TSSs and enhancers, supporting the presence of coordinated regulation within spatially organized domains.

A total of 73 stable TSS–enhancer interactions common across all tissues were identified, of which 21 corresponded to annotated genes. The remaining interactions may represent previously uncharacterized transcripts, non-coding RNAs, or tissue-specific isoforms not included in current annotations.

Functional analysis revealed that these genes are involved in transcription regulation, RNA metabolism, and nitrogenous base synthesis (Figure 3). This reflects a reprogramming of core cellular processes aimed at enhancing RNA and protein synthesis, which are essential for active growth and adaptation. Additionally, non-canonical Wnt signaling pathways, known to regulate cell proliferation and differentiation, were identified, suggesting their potential role in the development of muscle and other rapidly growing tissues.

These processes underpin the maintenance of cellular transcriptional activity and provide functional flexibility to regulatory networks. Therefore, the data suggest that positively correlated enhancer–promoter interactions facilitate the fine-tuned modulation of transcriptional levels rather than abrupt changes in gene expression.

Similar patterns have been reported in the FANTOM5 and ENCODE projects, where active enhancers are frequently associated with genes encoding transcription factors, cofactors, and components of the RNA-processing machinery [13,17]. This observation aligns with the concept of modular regulation, according to which active TADs constitute functional domains within which transcription is coordinated in spatially confined chromatin regions. Within these domains, genes involved in shared regulatory or metabolic pathways are synchronously activated, ensuring the stability and reproducibility of transcriptional programs.

It is noteworthy that among positively correlated (“++”) interactions, there was an enrichment of genes involved in the non-canonical Wnt signaling pathway, which plays a critical role in regulating growth, differentiation, and metabolism in avian species [18]. The involvement of the Wnt pathway in these interactions suggests that enhancer–promoter coactivation contributes to coordinating both the energetic and morphogenetic potential, particularly within the context of growth and metabolic adaptation.

In the majority of examined tissues, a positive correlation was observed between enhancer and promoter activity within individual TADs, with interactions of the “++” type (upregulated TSS/upregulated enhancer) predominating. The prevalence of “++” pairs suggests the functional coupling of regulatory elements and indicates that the activation of local enhancer clusters amplifies promoter expression within the same domain. This finding aligns with the model whereby transcriptional activation primarily occurs within a confined three-dimensional space formed by chromatin loops and stabilized by cohesin- and CTCF-dependent contacts [19,20], which restrict cis-regulatory element interactions. The highest density of such interactions was detected in the liver and brain—organs central to energy metabolism and plasticity.

KEGG pathway analyses further confirmed that these interactions participate in regulating transcription, RNA transport, splicing, and protein processing within the endoplasmic reticulum, reflecting an active reorganization of expression networks that support elevated metabolic and biosynthetic activity. Therefore, enhancer and promoter coactivation may represent an epigenetic adaptation mechanism that ensures the coordinated activation of genes essential for maintaining RNA homeostasis and protein synthesis (Figure 4).

The obtained results demonstrate that the interaction between enhancers and promoters in the chicken genome forms structurally organized regulatory modules within topologically associating domains (TADs). This spatial organization of transcriptional activity confirms that epigenetic regulation in birds follows the same fundamental principles as in mammals, where TADs are regarded as functional units coordinating gene expression by integrating promoters, enhancers, and their associated protein complexes [8,10,14]. The use of TAD maps constructed from Hi-C data obtained from blood cells and fibroblasts enabled the reconstruction of the cis-regulatory interaction network and the identification of TSS–enhancer pairs exhibiting coordinated expression changes between fast- and slow-growing samples.

### 2.3. Interaction Between Promoters and Enhancers on the TAD Level

We identified a statistically significant association between the expression of enhancers and promoters, aggregated over the same topologically associating domain (TAD). Despite a high level of noise, a significant correlation was observed between the combined expression levels of promoters and enhancers (Table 3). Notably, the median-based Theil–Sen regression model outperformed classical regression across all tissues, likely because a subset of TADs formed a subgroup in which the total promoter expression did not correlate with the total enhancer expression (Figure 5).

Despite the high statistical significance of the correlation coefficients and the strong regression values, we observed substantial variability in promoter expression relative to enhancer expression. Nevertheless, there was a clear trend indicating a relationship between promoter and enhancer expression within the same topologically associating domain (TAD). This behavior can be explained by the fact that genes located within a single TAD often belong to functionally related signaling pathways. Indeed, previous studies have demonstrated that genes within the same TAD tend to encode proteins with similar functions [21] and exhibit coordinated expression patterns [22].

Notably, promoter expression variability was higher than enhancer expression variability, which can be explained by the fact that a single gene is regulated by multiple enhancers within its TAD [23]. Additionally, gene expression variation was found to be associated with the expression of their regulators, including both enhancers and transcription factors [24].

### 2.4. Interaction Between Enhancers and Promoters on the TAD Level in the IsoSeq Experiment on Piao Chickens

We cross-validated our observation of statistical association between the enhancer and promoters that we predicted from the CAGE experiment on the F2 cross between Russian White and Cornish with the public data of the IsoSeq experiment of the Piao chickens (BioProject PRJNA961225, SRA ID SRR24293230). We also detected a statistically significant correlation (Table 4) and high levels of regression (Figure 6a) between the expression of promoters and enhancers, aggregated over the same TADs.

In order to validate the significance of the TAD boundaries, we permutated the IsoSeq dataset so that the aggregate expression of enhancers from each TAD was correlated against the aggregated expression of the promoters from the next adjacent TAD (Table 4 and Figure 6b). Expectantly, the correlations between enhancer and promoters from adjacent TADs were insignificant, and regression was correspondingly low.

Although the correlations were statistically significant and clear positive coactivation is evident in many TADs, considerable variability was observed: A subset of domains exhibits no correlation between the cumulative activity of promoters and enhancers (Figure 5 and Figure 6). This heterogeneity could stem from several biological and technical factors: (i) the presence of TADs dominated by repressors or insulator elements, (ii) tissue-specific reconfigurations of sub-TAD structure causing some enhancers to function independently of the promoters considered, (iii) post-transcriptional and transcriptional mechanisms not detectable through eRNA/CAGE levels (e.g., differences in processing rates or mRNA stability), and (iv) methodological limitations such as data noise, annotation inconsistencies, and differences in genome assemblies and mapping. The existence of a subgroup of TADs with disrupted correlation could explain why the Theil–Sen regression slope exceeded the classical SLR line: Theil–Sen is known to be more robust to outliers and subgroups.

Additionally, an asymmetry in variability was observed: Promoter expression exhibited greater variability than enhancer expression. This finding can be explained by several interconnected mechanisms. A single promoter may integrate signals from multiple enhancers, providing both redundancy and robustness in regulation. At the same time, the relative activity of individual promoters is sensitive to changes in chromatin accessibility, recruitment of transcription factors and cofactors, and 5’-UTR structure and alternative TSS usage. The increased variability of promoters emphasizes that regulation at the level of transcription initiation is more dynamic and context-dependent than enhancer transcriptional activity, which is reflected by eRNA levels.

The insignificant correlation and low regression between enhancers and promoters from different (not even distant) TADs, compared to the significant correlation between enhancers and promoters from the same TAD, suggest that our consensus TADs refelct, at least to seom extent, the real boundaries of chromotin contacts in the chicken genome.

### 2.5. Interactions Between Enhancers and Promoters in Chicken Brain

Brain tissue was the most enriched, with putative interactions between enhancers and promoters, while muscle breast and leg muscle tissues had the lowest number of possible interactions between the enhancers and promoters (Figure 7; Table 5).

The region of chromosome 13 between positions 11,663,167 and 13,151,229 was one of the most enriched, with putative interactions between promoters and enhancers. The interactions affected the promoters of six genes—ATOX1, CANX, LARP1, MAML1, PDGFRB, and SPARC (Figure 8).

All of the six genes could directly or indirectly affect growth rate in the studied animals. While CANX is related to autophagy and cell proliferation [25], MAML1 and PDGFRB influence the signaling pathways tied to adipogenesis and tissue growth. ATOX1, LARP1, and SPARC contribute through roles in metal homeostasis, protein synthesis, and matrix remodeling, respectively, which could indirectly affect body mass and metabolism.

## 3. Discussion

The interactions identified and the predominant positive regulation observed in fast-growing chickens suggest a comprehensive remodeling of regulatory networks that facilitate accelerated growth. This likely involves the activation of additional enhancers and modifications of promoter architecture as part of an epigenetic program designed to enhance the expression of metabolically active and proliferative genes. These findings emphasize the value of high-resolution models such as CAGE and highlight the necessity for further validation using complementary data sources, including ATAC-seq and histone mark ChIP-seq.

Our data illustrate how transcription factors, cofactors, transcription start sites (TSSs), and enhancers operate as coordinated modules structurally organized within topologically associating domains (TADs). The observed changes in promoter cluster width in fast-growing birds may reflect altered DNA accessibility, including the unveiling of previously repressed sites, as well as the recruitment of specific transcription factors and cofactors to promoter regions. Consequently, the transcriptional plasticity seen in fast-growing chickens—characterized by dynamic TSS usage, promoter width variation, and enhancer activation—forms a flexible and adaptive regulatory network integrated within the three-dimensional chromatin architecture. This plasticity likely plays a crucial role in supporting rapid tissue growth and development.

Our computational approach demonstrated statistical associations that allow us to propose the presence of biologically relevant interactions between promoters and enhancers in the chicken genome. Nevertheless, further experimental validations of the assumptions made from the dry-lab analysis of the data are undoubtedly required. 

## 4. Materials and Methods

### 4.1. Public CAGE and IsoSeq Data

Publicly available CAGE sequencing data from the “Genetic Technologies in Poultry” project http://chicken.biouml.org (last accessed on 20 November 2025) were utilized in this study: http://chicken.biouml.org/downloads/ChickenResearch2023/CageSeq/raw_data. (last accessed on 20 November 2025). The experiments involved 12 fast-growing and 12 slow-growing F2-generation chickens derived from a cross between the Russian White and Cornish breeds. Samples were collected from six tissues—brain, breast, heart, kidney, and legs—at 9 weeks of age. To validate the results, IsoSeq data from the NCBI SRA database (SRA ID SRR24293230) were used.

### 4.2. Expression of Promoters and Enhancers

Based on the results obtained in our previous study [26], we aligned CAGE reads from BGI-SEQ platform experiments to the GRCg7b reference genome using STAR aligner version 2.7.11b [27]. The chicken reference genome assembly RefSeq GRCg7b (RefSeq ID GCF_016699485.2) along with the corresponding chicken genome annotation release 106 from NCBI Gallus gallus was used as the reference.

The resulting alignments were then utilized to predict enhancers from the CAGE data by applying the bi-directional promoter detection approach [13], implemented in the CAGEr package version 2.8.0 [28], which serves as an interface to the quickEnhancers function from the CAGEfightR version 1.30.0 package [29].

We utilized genomic intervals of promoter regions predicted in the chicken genome GRCg7b using a method analogous to that described in our previous study [26]. Data were grouped by tissue, and CAGE tag clusters were constructed, resulting in one track for each of the six tissues. The resulting genomic intervals of the CAGE tag clusters were merged into a single set of promoter region intervals using the bedMerge tool of the BEDtools package.

Promoter and enhancer expression levels were quantified by counting reads aligned to the corresponding genomic intervals using the featureCounts package [30]. Differentially expressed promoters and enhancers were identified using the DESeq2 package [31].

### 4.3. Co-Localization of Promoters and Enhancers in Topologically Associating Domains

We determined the association of the predicted enhancers with topologically associating domains (TADs) in the chicken genome. As a reference TAD set, we used overlaps between JuiceBox2D annotated genomic intervals of fibroblast and erythrocyte TADs from the Ontogen database [8] (see also Appendix A). The TAD coordinates were converted from the original galGal5 assembly to the GRCg7b assembly using the liftOver tool [32].

Following coordinate conversion, three levels of spatial association between enhancer and promoter intervals were performed:Linear Proximity: Identifying enhancer–TSS/gene pairs located within 500,000 base pairs (bp) of each other.Co-Localization Within the Same TAD: Extracting all regulatory element–TSS/gene pairs that co-localize within the same TAD domain.Functional Filtering: From the set of interactions, only pairs in which both the enhancer and the corresponding TSS or gene were differentially expressed were retained.

Besides the restrictions on spatial association, we also excluded promoters and enhancers not differentially expressed between slow- and fast-growing samples. Thus, only significantly (Benjamini–Hochberg FDR lower than 0.05) up- or downregulated promoters and enhancers were considered.

We considered all enhancer–promoter pairs satisfying the above conditions as interacting.

### 4.4. Tissue-Wise Quantification of Interactions Between Promoters and Enhancers

We calculated all possible interactions between enhancers and promoters within the same TAD, restricting them to differentially expressed enhancers and promoters. Thus, the total number of promoter–enhancer interactions within a TAD was a mere product of the differentially expressed promoters and enhancers. Given a particular tissue, we calculated the sum of such products over all TADs. That number gave us the total number of promoter–enhancer interactions for that specific tissue. Since differential expression can be in the form of upregulation (+) or downregulation (−), we could annotate our hypothetical interactions as +/+, +/−, −/+, and +/− by separately calculating the products of up(down)regulated promoters and enhancers, correspondingly. By normalizing the total number of interactions within each tissue to 1.0, we calculated the proportions of interactions of different types (+/+, +/−, −/+, and +/−).

### 4.5. Correlation Between Aggregated Expression of Promoters and Enhancers in TADs

The genomic intervals of promoter regions were taken from our previous study [26], and the genomic intervals of enhancers were calculated using Andersson’s approach, as described above.

To quantify the expression of promoters and enhancers in CAGE-seq samples, we used the featureCounts tool. Then, the obtained counts were summarized TADs with the use of the bedSum tool of the BEDtools package.

To quantify the expression of previously promoters and enhancers in the IsoSeq experiment, we overlapped the IsoSeq reads against enhancer and promoter genomic intervals, extracting only the reads that overlapped promoters or enhancers, and we separately (using BEDtools intersect command with -u option) and subsequently calculated the coverage (in counts over intervals) of the overlapping reads over the intersection of erythrocyte and fibroblast TADs from [8] using the BEDtools coverage command.

All read counts were subsequently log transformed to base 2.

To assess and visualize the regression between the aggregated expression of promoters and enhancers, we used both simple linear regression (SLR) and the robust Theil–Sen estimator along with the Pearson, Speraman, and Kendall correlation coefficients.

## Figures and Tables

**Figure 1 ijms-26-11407-f001:**
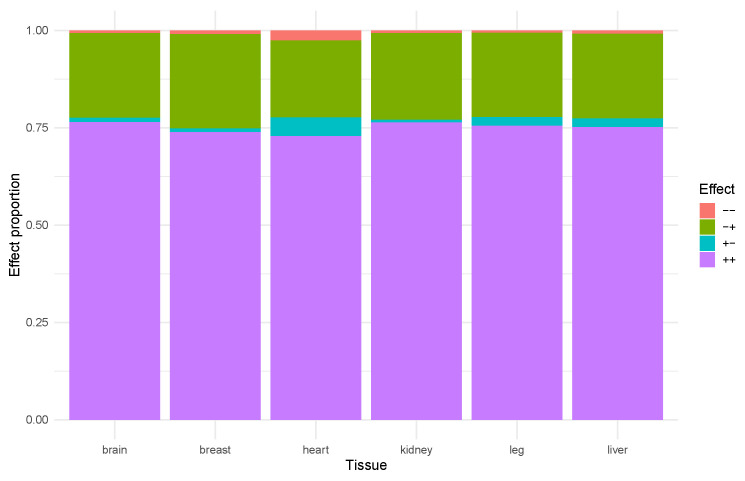
The relationship of TAD effects across different tissues. − −: Downregulated TSS and downregulated enhancers (shown in red); − +: downregulated TSS and upregulated enhancers (shown in green); + −: upregulated TSS and downregulated enhancers (shown in cyan); + + upregulated TSS and upregulated enhancers (shown in purple).

**Figure 2 ijms-26-11407-f002:**
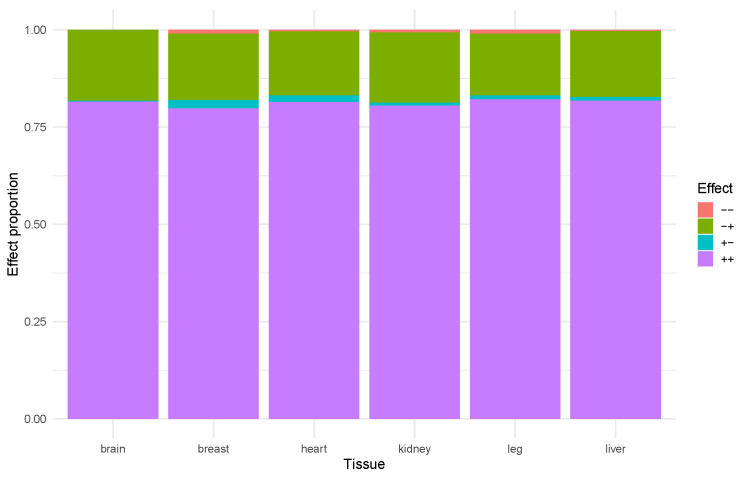
The relationship of consolidated TAD effects across different tissues. − −: Downregulated TSS and downregulated enhancers (shown in red); − +: downregulated TSS and upregulated enhancers (shown in green); + −: upregulated TSS and downregulated enhancers (shown in cyan); + +: upregulated TSS and upregulated enhancers (shown in purple).

**Figure 3 ijms-26-11407-f003:**
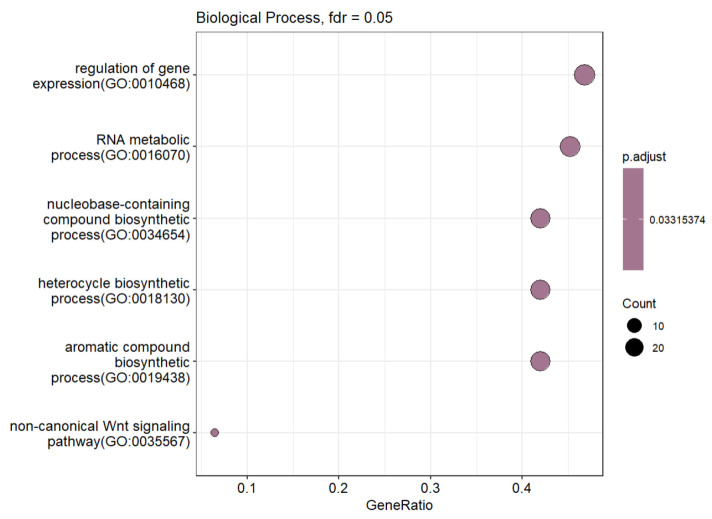
Over-representation analysis of genes with promoters that demonstrated correlation with enhancers within their TADs in all tissues.

**Figure 4 ijms-26-11407-f004:**
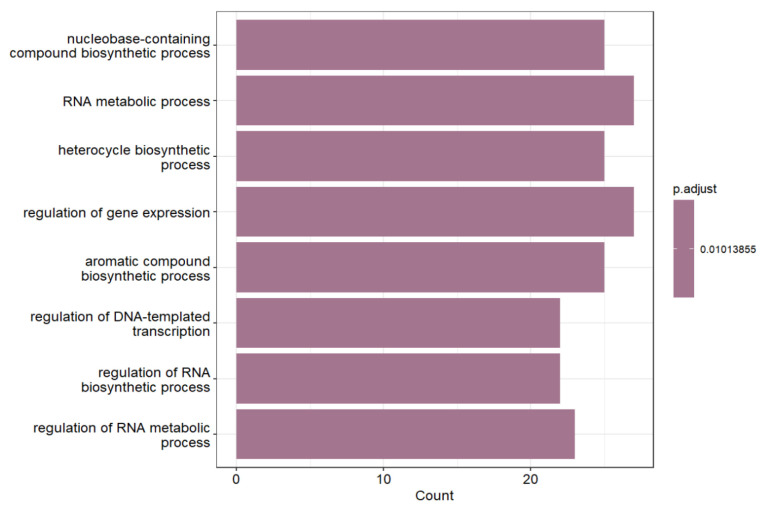
KEGG pathways over-represented in genes, where promoters participated in “++” interactions with enhancers within the same TAD.

**Figure 5 ijms-26-11407-f005:**
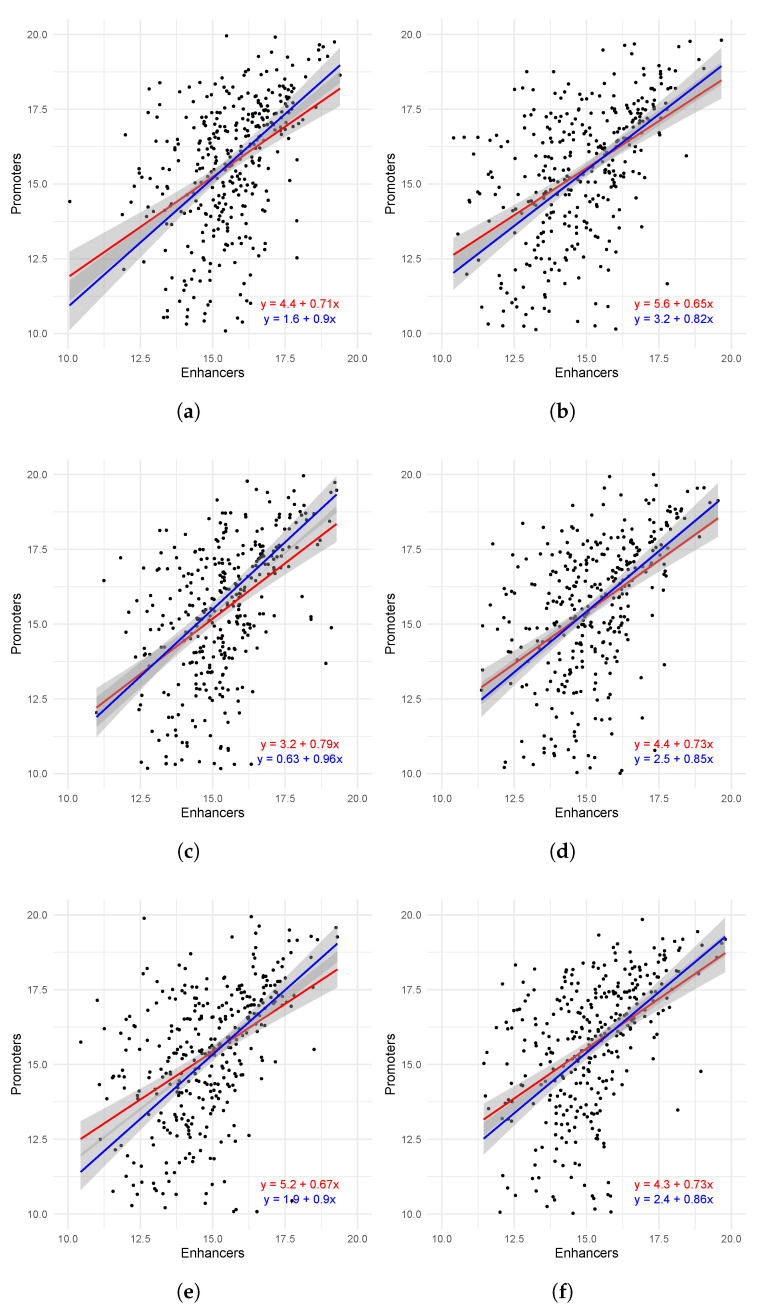
Linear regression between expression (in read counts) of enhancers and promoters aggregated over each TAD, in brain (**a**), breast (**b**), heart (**c**), kidney (**d**), leg (**e**), and liver (**f**). Each black dot represents aggregated expression of enhancers (X-axis) and promoters (Y-axis) over a single TAD. All axes are in logarithmic (base 2) scale. Simple linear regression (SLR) is shown in red; Theil–Sen regression is shown in blue. Boundaries of standard error are shaded in gray. (Also see Appendix B).

**Figure 6 ijms-26-11407-f006:**
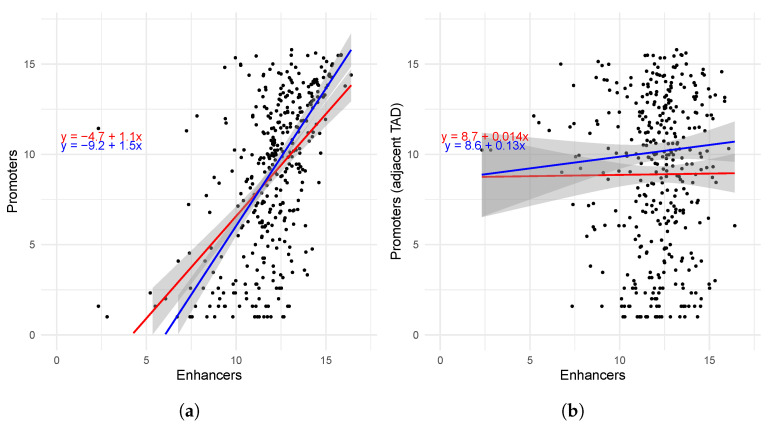
Linear regression between the expression (in read counts) of enhancers and promoters aggregated over each TAD in the tail tissues of Piao chickens: within the same TAD (**a**) and between the enhancer of a TAD and the promoter of the next TAD (**b**). Each black dot represents aggregated expression of enhancers (X-axis) and promoters (Y-axis) over a single TAD. All axes are in logarithmic (base 2) scale. Simple linear regression (SLR) is shown in red; Theil–Sen regression is shown in blue. Boundaries of standard error are shaded in gray. Also see Appendix B.

**Figure 7 ijms-26-11407-f007:**
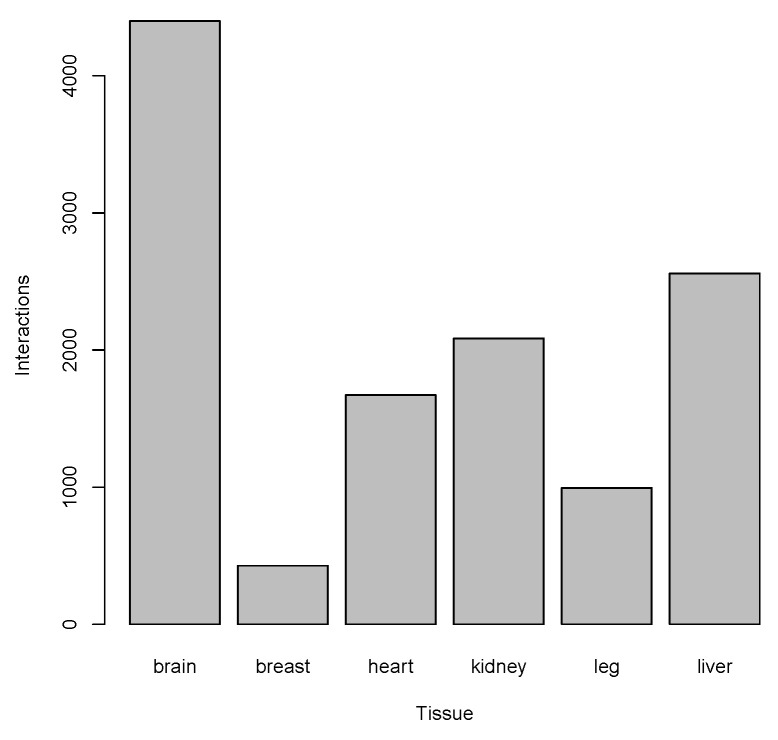
Number of putative enhancer–promoter interactions in tissues.

**Figure 8 ijms-26-11407-f008:**
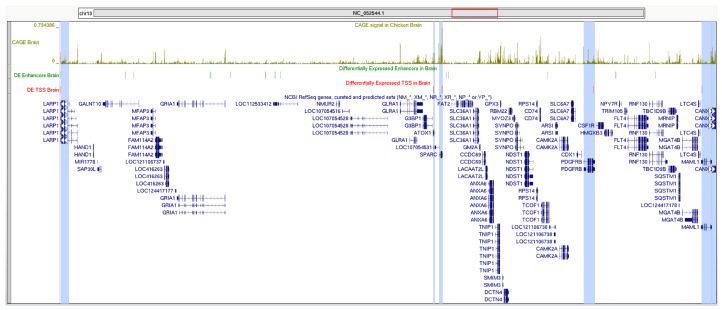
Region of chromosome 13 (genome position chr13:11,663,067-13,151,329), demonstrating multiple enhancers and proposedly interacting promoters of genes ATOX1, CANX, LARP1, MAML1, PDGFRB, and SPARC in brain tissues. Differentially expressed TSSs of the promoters are shown in red; blue vertical lines highlight location on the relevant RefSeq transcripts (shown in deep blue). Differentially expressed enhancers are shown in dark green. Aggregated CAGE signal in brain samples is shown in olive. Location of the region on the chromosome 13 (NCBI ID NC_052544.1) is shown with red rectangle.

**Table 1 ijms-26-11407-t001:** Number of intervals in original and lifted-over TAD sets from Ontogen, their overlap with the Jaccard statistic, and the number of tissue-specific TADs.

TADs	Genome	#Intervals	Jaccard
Erythrocyte	galGal5	507	
Erythrocyte	GRCg7b	515	
Fibroblast	galGal5	403	
Fibroblast	GRCg7b	420	
Erythrocyte × Fibroblast	GRCg7b	769	0.780
Erythrocyte only	GRCg7b	9	
Fibroblast only	GRCg7b	14	

**Table 2 ijms-26-11407-t002:** Regulation of TFs and cofactors by enhancers within TADs.

Tissue	Effect	Count	++	−+	+−	− −
Brain	TFs	31	108	2		
	Cofactors	23	51	4		
Breast	TFs	9	15			
	Cofactors	7	7	1		
Heart	TFs	20	42		3	
	Cofactors	19	31	2	2	1
Kidney	TFs	26	70		1	
	Cofactors	17	29	2		
Liver	TFs	30	91			
	Cofactors	23	41	4	1	
Leg	TFs	17	30			
	Cofactors	13	19			

**Table 3 ijms-26-11407-t003:** Correlation between expression of enhancers and promoters aggregated over each TAD. Pearson’s *r*, Spearman’s ρ, and Kendall’s τ are given, along with the statistical significance of them not being equal to zero.

Tissue	*r*	P(r≠0)	ρ	P(ρ≠0)	τ	P(τ≠0)
Brain	0.437	3.38 × 10^−18^	0.489	5.93 × 10^−23^	0.343	2.88 × 10^−22^
Breast	0.457	5.84 × 10^−20^	0.506	8.8 × 10^−25^	0.360	2.52 × 10^−24^
Heart	0.482	2.73 × 10^−22^	0.515	0.0	0.362	1.47 × 10^−24^
Kidney	0.473	2.16 × 10^−21^	0.513	1.68 × 10^−25^	0.365	5.17 × 10^−25^
Leg	0.461	2.84 × 10^−20^	0.509	0.0	0.363	9.85 × 10^−25^
Liver	0.489	6.12 × 10^−23^	0.532	0.0	0.377	1.32 × 10^−26^

**Table 4 ijms-26-11407-t004:** Correlation between expression of enhancers and promoters aggregated over each TAD in Piao chicken tail tissue. Correlations within the same TAD and between enhancers of one TAD and promoters from the adjacent TAD are shown. Pearson’s *r*, Spearman’s ρ, and Kendall’s τ are given, along with the statistical significance of them not being equal to zero.

TADs	*r*	P(r≠0)	ρ	P(ρ≠0)	τ	P(τ≠0)
Same TAD	0.534	9.13 × 10^−29^	0.551	6.43 × 10^−31^	0.400	1.66 × 10^−30^
Adjacent TADs	0.00662	0.899	0.0487	0.35	0.0323	0.354

**Table 5 ijms-26-11407-t005:** Number of putative enhancer–promoter interactions in tissues as shown in Figure 7.

Tissue	#Interactions
Brain	4400
Breast	428
Heart	1672
Kidney	2085
Leg	995
Liver	2559

## Data Availability

The original contributions presented in this study are included in the article/Appendix B. Further inquiries can be directed to the corresponding author.

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
