# Peer review of "Interaction Between Enhancers and Promoters in Chicken Genome"

_ijms, 2025, doi:10.3390/ijms262311407_

Round 1

Reviewer 1 Report

Comments and Suggestions for Authors

In the reviewed study, Grushina et al. analyzed dependence between activities of genes and enhancers located within the same topologically-associated domains (TADs). More precisely, the Authors analyzed the dependence between the changes of the transcriptional activities of genes and the changes of the activites of enhancers, between slow- and fast-growing chicken from F2 generation of the cross between two chicken breeds. The presented analysis is based on publicly available CAGE sequencing data, which the Authors processed using established methods, to obtain expression of genes and location and activity of enhancers. The analyses were performed separately based on CAGE expression data for six organs (brain, brest, heart, kidney, leg, and liver).

The Authors used published coordinates of TAD boundaries in embryonic chicken fibroblasts and in erythrocytes, to identify genes, identified by transcription start sites (TSSs), and enhancers located within the same TADs. Additionally, the authors applied a distance limit and filtering of both genes and enhancers with changed activity between the slow- and fast-growing chicken. The research question is interesting and the Authors are clearly proficient in analyzing the data.

1. However, the description of the analysis steps, in particular of enhancer-promoter mapping and data aggregation, and of the content of figures and tables content is insufficient. As consequence, in several places it is impossible to understand with certainty, what was done, and what is represented in the figures. This must be corrected.

2. The Authors seemingly use only one way of aggregating promoters and enhancers, and this way utilizes both the TAD boundaries and the linear proximity. The work would be more interesting, if the results obtained using the TAD boundaries were compared with the results obtained using the same distribution of linear distances across the TAD boundaries.

The detailed questions and comments:

Ad 1.

Line 54: How was the IsoSeq data used to validate the results?

Lines 85-86: “the enhancer and the corresponding TSS or gene” – How was the correspondence (mapping) between the promoters/TSSs and enhancers performed? All-to-all in the same TAD within the linear distance limit, or otherwise, or not at all? Please describe under Material and Methods.

Lines 93-94: “experimentally defined TAD regions […] were incorporated into analysis” – The web site accompanying the reference [15] (Ontogen database) contains several TAD files, generated with 3 different algorithms. Indicate, which files were used in the current work.

Lines 105-108: “merged into conditional clusters or regulatory domains” – provide more details, permitting understanding what was done.

Lines 116-117 There appears to be a discrepancy between the main text, which reads: “Specifically, most cases were characterized by either synchronous activation (Log2FC > 0 in both enhancers and TSS) or a simultaneous downregulation.” and the caption/legend of Figure 1 and 2, in which the top fraction is violet (++), in agreement with the main text, but the second largest fraction is green (−+), not orange (−−).

Figure 1 and Figure 2: The proportions of what are shown in these figures? Of TADs? Of promoter-enhancer pairs?

Line 120, 121: Explain what is meant by “coordinated response to enhancer activity”?

Table 1: What in meant by Interactions in this table?

The data shown in Table 2 and in Figure 4 are potentially very interesting. However, the present description of their content is confusing. The main text reads in lines 150-151: “a significant correlation was observed between the combined expression levels of promoters and enhancers”, suggesting aggregation of the expression in every TAD, separately for enhancers and promoters, and then computing correlations of two vectors of aggregated values indexed by TADs, or the regression analysis between the two expression values for every TAD. The caption of Table 2 however reads: “Correlation between enhancers and promoters within a TAD.” and also the caption of Figure 4 reads: “Linear regression between total expression of enhancers and promoters within a single TAD.” It is also not clear, what is meant by “total expression levels” in Line 148 and the caption of Figure 4 and “expression” in the caption of Table 2. In particular is it (log) fold change or – please clarify.

Ad 2.

The statement in the discussion “Our data indicate that transcription factors, cofactors, transcription start sites (TSS) and enhancers operate as coordinated modules structurally organized within topologically associated domains (TADs).” is not sufficiently supported by the reported analyses.
To justify the above statement the Authors should either: (A) (preferably) compare the current results, obtained using the TAD boundaries, with the results obtained without using the TAD boundaries, for example, by repeating the key analyses (Table 2 and/or Fig. 4), using the center of every TAD instead of the its boundaries; or (B) reduce the claim, for example, by changing “indicate that” to “illustrate how”.

Author Response

Comments Ad.1 Line 54: How was the IsoSeq data used to validate the results?
Response: We added the missing part on the IsoSeq validation. We estimated the correlation and regression between expression of the predicted enhancers and promoters similarly to Table 2 and Figure 4. We described the process in Materials and Methods, subsection “Correlation between aggregated expression of promoters and enhancers in TADs”, lines 116 – 125. We discussed the results in subsection “Interaction between enhancers and promoters on the TAD level in the IsoSeq experiment on Piao chickens”, lines 281 – 288, and added Table 4 and Figure 6.

Comments Ad.1 Lines 85-86: “the enhancer and the corresponding TSS or gene” – How was the correspondence (mapping) between the promoters/TSSs and enhancers performed? All-to-all in the same TAD within the linear distance limit, or otherwise, or not at all? Please describe under Material and Methods.
Response: In our study we hypothesized that an enhancer could regulate all promoters within the same TAD, and relied our enhancer-promoter mapping on that notion. Thus, it is all-in-all in the same TAD. Besides that, we only considered differentially expressed enhancers and promoters. We have clarified that in Materials and Methods, subsection “Co-localization of promoters and enhancers in topologically associating domains” lines 83 – 84, 91 – 94 and 95 – 96.

Comments Ad.1 Lines 93-94: “experimentally defined TAD regions […] were incorporated into analysis” – The web site accompanying the reference [15] (Ontogen database) contains several TAD files, generated with 3 different algorithms. Indicate which files were used in the current work.
Response: We have described how we merged the fibroblast and erythrocyte datasets from the Ontogen database into one set of intervals (Materials and Methods, lines 79 – 82). Accordingly, we have rephrased the mentioned lines in the Results, into more clear wording (subsection “Enhancer-mediated epigenetic regulation of promoter activity” lines 161 – 162)

Comments Ad.1 Lines 116-117:
There appears to be a discrepancy between the main text, which reads: “Specifically, most cases were characterized by either synchronous activation (Log2FC > 0 in both enhancers and TSS) or a simultaneous downregulation.” and the caption/legend of Figure 1 and 2, in which the top fraction is violet (++), in agreement with the main text, but the second largest fraction is green (−+), not orange (−−).
Response: We agree that the interpretation of the results is misleading to a reader in the current form. Indeed, in most cases we observed the increase of enhancer expression along with either downregulation or upregulation of promoters. The word ‘simultaneous’ was used to indicate that such behavior was observed in all differentially expressed enhancers and promoters located within the relevant TAD. We have rephrased the relevant part of Results to clarify, (subsection “Enhancer-mediated epigenetic regulation of promoter activity”, lines 172 – 178)

Comments Ad.1 Figure 1 and Figure 2: The proportions of what are shown in these figures? Of TADs? Of promoter-enhancer pairs?
Response: We calculated all possible interactions between enhancers and promoters within the same TAD, restricting them to differentially expressed enhancers and promoters. Thus, the total number of promoter-enhancer interactions within a TAD was a mere product of the differentially expressed promoters and enhancers. Given a particular tissue, we calculated the sum of such products over all TADs. That number gave us the total number of promoter-enhancer interactions for that specific tissue. Since differential expression can be upregulation (+) or downregulation (-), we could annotate our hypothetical interactions as +/+, +/-, -/+, and +/-, by separately calculating the products of up(down)-regulated promoters and enhancers, correspondingly. By normalizing the total number of interactions within each tissue to 1.0, we calculated the proportions of interactions of different types and plotted them in Figures 1 & 2. We have explained the way we calculated the number and proportion of interactions in Materials and Methods, subsection “Tissue-wise quantification of interactions between promoters and enhancers” (lines 97 – 108)

Comments Ad.1 Line 120, 121: Explain what is meant by “coordinated response to enhancer activity”?
Response: We analysed the effect of promoters which were located within a single TAD and demonstrated significant change in activity being all up-regulated or down-regulated. We have rephrased the mentioned part of the Results, subsection “Enhancer-mediated epigenetic regulation of promoter activity”, to clarify (lines 180 – 182).We have also specified in Materials and Methods that we considered only differentially expressed promoters and enhancers (subsection “Co-localization of promoters and enhancers in topologically associating domains”, lines 91 – 94)

Comments Ad.1 Table 1: What is meant by Interactions in this table?
Response: The heading of Table 1 was wrong and ‘interactions’ were introduced by mistake. The column under question is not ‘interactions’, it is count of differentially expressed promoters within all TADs annotated as promoters of TFs or promoters of TF cofactors, which hypothetically participate in cis-interactions with differentially expressed enhancers within the same TAD. We have modified the former Table 1 (now it is Table 2) and its caption to clarify (lines 205 – 206).

Comments Ad.1 Table 1 and Figure 4: The data shown in Table 2 and in Figure 4 are potentially very interesting. However, the present description of their content is confusing. The main text reads in lines 150-151: “a significant correlation was observed between the combined expression levels of promoters and enhancers”, suggesting aggregation of the expression in every TAD, separately for enhancers and promoters, and then computing correlations of two vectors of aggregated values indexed by TADs, or the regression analysis between the two expression values for every TAD. The caption of Table 2 however reads: “Correlation between enhancers and promoters within a TAD.” And also the caption of Figure 4 reads: “Linear regression between total expression of enhancers and promoters within a single TAD.” It is also not clear what is meant by “total expression levels” in Line 148 and the caption of Figure 4 and “expression” in the caption of Table 2. In particular is it (log) fold change or – please clarify.
Response: We agree that the captions in Table 2 and Figure 4 are misleading in the current form and do not reflect what was really done. Indeed, we separately aggregated expression of enhancers and promoters by TAD, and then calculated correlation and regression between two vectors. We have corrected the captions of the former Table 2 (now it is Table 3) and the former Figure 4 (now it is Figure 5) accordingly (Results, lines 267 – 268). We agree that the term ‘total expression’ is not exact and does not reflect what was correlated in Figure 4. In this case, we meant aggregated expression of promoters and enhancers over TADs. We have changed the wording in the caption of the former Figure 4 (now it is Figure 5) (Results,  lines 267 – 268)  and added the description of the correlation and linear regression performed in the study (Materials and Methods, Subsection “Correlation between aggregated expression of promoters and enhancers in TADs” lines 110 – 115) We used log2 transformed read counts for the plots and correlation estimates, we added this in Materials and Methods (line 122) and to the caption of the Figure 4 (lines 267 – 268). We also rephrased the unclear term “total expression” in line 148 with more relevant wording “expression of enhancers and promoters, aggregated over the same topologically associating domain (TAD)” (Results, subsection “Interaction between enhancers and promoters within a single TAD”, lines 261 – 262)

Comments Ad 2. The statement in the discussion “Our data indicate that transcription factors, cofactors, transcription start sites (TSS) and enhancers operate as coordinated modules structurally organized within topologically associated domains (TADs).” is not sufficiently supported by the reported analyses.
To justify the above statement the Authors should either: (A) (preferably) compare the current results, obtained using the TAD boundaries, with the results obtained without using the TAD boundaries, for example, by repeating the key analyses (Table 2 and/or Fig. 4), using the center of every TAD instead of the its boundaries; or (B) reduce the claim, for example, by changing “indicate that” to “illustrate how”.
Response: We have rephrased our conclusion to softer wording: “Our data illustrate how transcription factors, cofactors, transcription start sites (TSS), and enhancers operate as coordinated modules structurally organized within topologically associating domains (TADs).” (line 342). Additionally, we have validated the TAD boundaries by correlating the aggregated expression of enhancers by TADs against the aggregated expression of promoters over the adjacent TADs and observed how correlation dropped to insignificant level. We described this in Results, subsection “Interaction between enhancers and promoters on the TAD level in the IsoSeq experiment on Piao chickens”, lines 289 – 293 and discussed in lines 315 – 319.

Reviewer 2 Report

Comments and Suggestions for Authors

The manuscript examined the correlation between gene and enhancer activities within individual TADs across multiple tissues in slow- and fast-growing chickens, and assessed the biological significance of genes whose promoters were regulated by enhancers. The study revealed a statistically significant association between gene expression levels and enhancer activity in all tissues examined. This work is commendable for integrating multi-tissue data and leveraging 3D genome organization to explore enhancer–gene regulation in a growth-related context, which represents a novel and timely contribution to the field. However, several major and minor issues need to be addressed before this manuscript is suitable for publication.

Major issues:

  1. This is a bioinformatics manuscript. The main results and conclusions of this study are entirely based on computational predictions and correlation analyses. These approaches are valuable for identifying potential interactions, but it is important to note that correlation does not imply caustation. Experimental validation is therefore needed. I strongly recommend that the authors supplementing their analysis with 3C (Chromosome Conformation Capture) or 3C-qPCR experiments. If experimental validation is not feasible, the authors should clearly state the limitations of the study and present their conclusions strictly as computational predictions.
  2. In the Materials and Methods section (Lines 53-54), the tissue “breast” is listed twice in the sentence “Samples were collected from six tissues—brain, breast, heart, kidney, legs, and breast—at 9 weeks of age.” This seems abnormal. Could the authors please clarify this?
  3. I found a significant methodological weakness in in this study. The authors analyze transcriptional regulation across six different tissues (Lines 53-54). However, the manuscript indicates that the TAD data employed originate from chicken blood cells and fibroblasts (Lines 103-104). Since three-dimensional genome organization is highly tissue-specific, using TAD datasets from non-matching tissues may affect the accuracy and interpretability of the results. This represents an important methodological limitation. The authors should address this major issue.
  4. At present, the manuscript reads more like a statistical analysis report rather than a study presenting a complete biological story. The functional enrichment analyses are rather broad, and the overall length and depth of the manuscript suggest that the biological insights have not been fully explored.

I suggest that the authors select a single “key TAD” most strongly associated with growth-related traits as a case study, and illustrate it in a new figure. This figure could include a genome browser view integrating genes, enhancers, and CAGE signals; functional analyses of key genes within the TAD (e.g., growth factors or myogenesis-related genes); and motif analysis of critical enhancer sequences. Such a case study would transform abstract statistical results into concrete biological mechanisms, enhancing both the depth and impact of the manuscript.

Minor issues:

  1. Line 13: The keyword “enhacers” is a spelling error, should be revised to “enhancers.”
  2. Lins 68-69: In the sentence “Data were grouped by tissue, and CAGE tag clusters were constructed, resulting in one track per each of the six tissues,” the words “per” and “each” should not be used together. Please revise.
  3. In the subtitle of section 3 (Results), “Enhancer-mediated epigenetic regulation regulation of promoter activity,” the word “regulation” appears twice. Please correct this.
  4. In the legends of Figures 1 and 2, the phrase “TSS dowregulated” should be corrected to “TSS downregulated.”

Author Response

Comments Major 1: This is a bioinformatics manuscript. The main results and conclusions of this study are entirely based on computational predictions and correlation analyses. These approaches are valuable for identifying potential interactions, but it is important to note that correlation does not imply causation. Experimental validation is therefore needed. I strongly recommend that the authors supplement their analysis with 3C (Chromosome Conformation Capture) or 3C-qPCR experiments. If experimental validation is not feasible, the authors should clearly state the limitations of the study and present their conclusions strictly as computational predictions.
Response: We agree that this study was purely bioinformatical and the relevant project did not suggest any experimental validation. We also agree that the theoretical nature of the study and hypothetical nature of the results obtained and  conclusions made require more pronounced disclaimer and detailed discussion. We have expanded the Introduction and Discussion sections accordingly (lines 47 – 48 and 352 – 365).

Comments Major 2: In the Materials and Methods section (Lines 53-54), the tissue “breast” is listed twice in the sentence “Samples were collected from six tissues—brain, breast, heart, kidney, legs, and breast—at 9 weeks of age.” This seems abnormal. Could the authors please clarify this?
Response: The second appearance of "breast" in the sentence was a typo and we have corrected it (lines 55-56).

Comments Major 3:
I found a significant methodological weakness in this study. The authors analyze transcriptional regulation across six different tissues (Lines 53-54). However, the manuscript indicates that the TAD data employed originate from chicken blood cells and fibroblasts (Lines 103-104). Since three-dimensional genome organization is highly tissue-specific, using TAD datasets from non-matching tissues may affect the accuracy and interpretability of the results. This represents an important methodological limitation. The authors should address this major issue.
Response: Indeed, we used the TAD from data on blood cells and distributed that results on other tissues. We agree that such induction looks questionable without detailed explanation and discussion. Nevertheless, we have compared the TADs from two different cell types – fibroblasts and erythrocytes. Although both cell types derive from the same bone marrow progenitor, they have totally different fate in hematopoiesis. We have noticed that both cell types share most of their TAD intervals and tissue-specific fragments of TADs are nonsignificant. We distributed the notion of similarity between TADs of different blood cells on other tissues. Thus, we implicitly made a hypothesis of TAD similarity between distantly related or non-related tissues. Actually, it is known that the substantial number of TADs is stable between tissues (https://doi.org/10.1016/j.ajhg.2021.01.001).  We agree that such a hypothesis has its limitations which must be explicitly claimed and discussed. We have added the proper explanations in Methods (lines 79 – 81) and Results (lines 127 – 143).

Comments Major 4: At present, the manuscript reads more like a statistical analysis report rather than a study presenting a complete biological story. The functional enrichment analyses are rather broad, and the overall length and depth of the manuscript suggest that the biological insights have not been fully explored.
Response: We added the interpretation of biological implications of our results as well as additional over-representation analysis (lines 188 – 195, 216 – 259, and 294 – 319).

Comments Major 5: I suggest that the authors select a single “key TAD” most strongly associated with growth-related traits as a case study, and illustrate it in a new figure. This figure could include a genome browser view integrating genes, enhancers, and CAGE signals; functional analyses of key genes within the TAD (e.g., growth factors or myogenesis-related genes); and motif analysis of critical enhancer sequences. Such a case study would transform abstract statistical results into concrete biological mechanisms, enhancing both the depth and impact of the manuscript.
Response: We found that the brain tissue was the most involved in interactions between promoters and enhancers, thus we found a TAD which was the richest in the enhancer-promoter interactions and analysed it in detail. We added a subsection “Interactions between enhancers and promoters in chicken brain” in Results (lines 320 – 332)

Comments Minor 1: Line 13: The keyword “enhacers” is a spelling error, should be revised to “enhancers.”
Response: We have corrected the typo (line 13).

Comments Minor 2: Lines 68-69: In the sentence “Data were grouped by tissue, and CAGE tag clusters were constructed, resulting in one track per each of the six tissues,” the words “per” and “each” should not be used together. Please revise.
Response: We have rephrased the sentence to better grammar and style (lines 69 –71).

Comments Minor 3: In the subtitle of section 3 (Results), “Enhancer-mediated epigenetic regulation regulation of promoter activity,” the word “regulation” appears twice. Please correct this.
Response: We have corrected the typo (line 144).

Comments Minor 4: In the legends of Figures 1 and 2, the phrase “TSS dowregulated” should be corrected to “TSS downregulated.”
Response: We have corrected the typos in the legends of the Figures 1 and 2 (lines 179 – 180 and 187 – 188)

Round 2

Reviewer 1 Report

Comments and Suggestions for Authors

The Authors have fully addressed my comments. I have also read with interest the additions made in response to the other Reviewer's comments. The improvements in the current revised version are substantial, with the addition of new figures with results that that fully substantiate the Authors original claims. The precision of description and the quality of presentation is now also fine.

Reviewer 2 Report

Comments and Suggestions for Authors

I recommend acceptance of this manuscript. The authors have addressed all major concerns, added solid data supporting their consensus TAD approach, and included a clear case study (Section 3.5, Figures 7–8) linking their results to biological mechanisms. The revision is strong and suitable for publication.